# Computed Tomography (CT) Image Quality Enhancement via a Uniform Framework Integrating Noise Estimation and Super-Resolution Networks

**DOI:** 10.3390/s19153348

**Published:** 2019-07-30

**Authors:** Jianning Chi, Yifei Zhang, Xiaosheng Yu, Ying Wang, Chengdong Wu

**Affiliations:** 1Faculty of Robot Science and Engineering, Northeastern University, Shenyang 110004, China; 2College of Information Science and Engineering, Northeastern University, Shenyang 110004, China

**Keywords:** medical image enhancement, deep convolutional neural network, dense connection, inception structure, residual network

## Abstract

Computed tomography (CT) imaging technology has been widely used to assist medical diagnosis in recent years. However, noise during the process of imaging, and data compression during the process of storage and transmission always interrupt the image quality, resulting in unreliable performance of the post-processing steps in the computer assisted diagnosis system (CADs), such as medical image segmentation, feature extraction, and medical image classification. Since the degradation of medical images typically appears as noise and low-resolution blurring, in this paper, we propose a uniform deep convolutional neural network (DCNN) framework to handle the de-noising and super-resolution of the CT image at the same time. The framework consists of two steps: Firstly, a dense-inception network integrating an inception structure and dense skip connection is proposed to estimate the noise level. The inception structure is used to extract the noise and blurring features with respect to multiple receptive fields, while the dense skip connection can reuse those extracted features and transfer them across the network. Secondly, a modified residual-dense network combined with joint loss is proposed to reconstruct the high-resolution image with low noise. The inception block is applied on each skip connection of the dense-residual network so that the structure features of the image are transferred through the network more than the noise and blurring features. Moreover, both the perceptual loss and the mean square error (MSE) loss are used to restrain the network, leading to better performance in the reconstruction of image edges and details. Our proposed network integrates the degradation estimation, noise removal, and image super-resolution in one uniform framework to enhance medical image quality. We apply our method to the Cancer Imaging Archive (TCIA) public dataset to evaluate its ability in medical image quality enhancement. The experimental results demonstrate that the proposed method outperforms the state-of-the-art methods on de-noising and super-resolution by providing higher peak signal to noise ratio (PSNR) and structure similarity index (SSIM) values.

## 1. Introduction

Computed tomography (CT) imaging technology has played an increasingly significant role in the computer assisted diagnosis system (CADS) [1] for health examination and disease detection. However, the digital image quality generated from recent imaging devices is always interrupted by noise, storage, and transmission loss [2], resulting in noisy low-resolution images that may degrade the effects of the subsequent steps of CADs, including segmentation, feature extraction, and diagnosis. Therefore, the image quality enhancement, especially image de-noising and image super-resolution, have drawn more and more attention of the researchers in the past two decades.

Given the fact that the degradation process of an image can be summarized as blurring [3], down-sampling [4], and noise interference [5], traditional image quality enhancement methods typically include two steps: Image de-noising was performed first and then the image super-resolution was implemented subsequently. Image de-noising methods can be categorized into two types: (1) Prior model-based algorithms and (2) deep learning-based algorithms. Prior model-based algorithms assume the noisy image is a function of the noise-free image and noise component and replace the values of “noise” pixels by the values estimated from those “noise-free” pixels following some models. Some representative model-based de-noising methods include the non-local means algorithm (NLM) [6], singular value decomposition algorithm (K-SVD) [7], and block-matching and 3D filtering (BM3D) [8]. However, it is difficult to establish a model that can differentiate noise and image details effectively since noise is non-uniformly distributed in medical images. Contrarily, deep learning-based algorithms take advantage of the “black box” structures [9], such as the convolutional neural network (CNN) model [10], multi-layer perception (MLP) model [11], and generative adversarial network (GAN) model [12]. These end-to-end methods simulate the information processing procedure of humans and learn the noise-free image from the noisy image through hierarchical network frameworks. However, they are still associated with over-smoothed edges and a loss of details since the structure information is likely lost when going through the network. For the super-resolution step, most widely used methods can be categorized into three types: (1) Interpolation-based methods, (2) reconstruction-based methods, and (3) learning-based methods. Bicubic [13] and cubic spline [14] are common means of interpolation-based methods. They can increase the resolution of the image efficiently but suffer from poor visual performance since they do not take the structure information into account. The maximum posterior probability-based method [15], the singular value decomposition-based method [16], and the non-local structure similarity based-method [17] are some typical reconstruction based methods, which can recover more details from the low-resolution image but require more prior knowledge about the degradation or the imaging environment. Similar to those methods for image de-noising, learning-based methods have drawn more and more attention in super-resolution because they integrate the patch extraction and aggregation, dictionary learning, and pre-processing in one uniform framework, such as the super-resolution convolutional network (SRCNN) [18], generative adversarial network for super-resolution (SRGAN) [19], efficient sub-pixel convolutional neural network (ESPCN) [20], and very deep convolutional networks (VDSR) [21].

The traditional “two-step” image quality enhancement methods discussed above cannot provide a convincing performance. The main reason is that de-noising is essentially a smoothing process, while super-resolution is contrarily a sharpening process, and it is difficult to unify them in framework. Zhang et al. [22] proposed the super-resolution network as a multiple degradation (SRMD) method by applying a uniform framework to perform de-noising and super-resolution simultaneously. They concatenated the noise level and blur kernel then stretched them into degradation maps and took the degraded image and degradation maps as the input of the convolutional neural network (CNN). The SRMD was the first attempt to unify noise, blurring, and down-sampling (low resolution) into one network so obviously improved the practicability. After that, more variants of SRMD were proposed for enhancement of image quality, such as the multi-image super resolution reconstruction (SRR) [23] and the residual-dense network for super-resolution (RDN) [24]. However, all these methods have the following shortcomings: (1) They are all non-blind models, which require the image noise level as prior information; (2) the CNN networks applied in the SRMD are quite shallow regarding efficiency but do not fully reflect the details and textures in the image; and (3) the mean square error (MSE) loss function used in recent networks is likely to lose a large amount of edge or structure information, resulting in over-smoothing.

Given to the above three problems of the existing SRMD networks, we took the following methods into account: (1) Noise estimation methods, (2) deeper super-resolution networks, and (3) comprehensive loss function. For noise estimation, classic methods that extracted noise from the image first and estimated the noise level as the statistical variance [25,26,27,28] are not suitable in medical images, where there are more high-frequency details than noise. In contrast, enlightened by the ability of CNNs to classify different types of images [29,30,31,32], we propose a novel image noise level classification network by incorporating inception structures and dense connections to make full use of their advantages in extracting global noise features. The inception structure [31] can extract the features of noise under multiple receptive fields and dense connections can reuse all the extracted features to ensure noise features transfer across the whole network. For the super-resolution network, we modified the dense-residual network [24] by integrating inception blocks. A residual-dense network can deepen the learning process without gradient vanishing. We embedded the inception block in each of the skip connection routines of the dense-residual network so that the structure features of the image are transferred through the network more than the noise and blurring features, leading to better effects of non-linear mapping from the degraded image to the high-quality image. Moreover, for the loss function, the perceptual loss and mean square error (MSE) loss are combined together, so both the pixel level evaluation and visual effect of the enhanced image can be further improved. Figure 1 shows the framework of our proposed image quality enhancement method. In short, we make the following contributions in this work:We propose a dense-inception network integrating an inception structure and dense skip connection to estimate the noise level. Note that we follow the definition of noise level in [25,26,27,28] as the variance of the Gaussian noise added in the image. To the best of our knowledge, it is the first attempt to adopt a deep convolutional neural network to estimate noise in medical images. Different from the traditional methods that need to extract noise from the image first, we train a network to classify a noisy medical image into the class labelled by the variance of the noise distribution.We propose a modified residual-dense network to reconstruct the high-resolution image with low noise. The inception block is applied on each skip connection of the dense-residual network so that the structure features of the image are transferred through the network more than the noise and blurring features. Moreover, the noise level that is needed for reconstruction is directly calculated by the proposed noise level estimation network, so the reconstruction network can process the noisy blurred medical images with much less prior knowledge about the noise than the state-of-the-art works.We propose a joint loss function to restrain the super-resolution network. Both the perceptual loss and the mean square error (MSE) loss are used to achieve a better performance in the reconstruction of image edges and details.

We applied our method to the Cancer Imaging Archive (TCIA) public dataset to evaluate its ability in medical image quality enhancement. The experimental results demonstrate that the proposed degradation estimation network achieves a higher accuracy in the classification of noise levels than comparators. After combining the degradation estimation and super-resolution network, the proposed method outperforms the state-of-the-art methods on de-noising and super-resolution by providing higher PSNR and SSIM values. The proposed method can jointly perform de-noising and super resolution to obtain a better visual effect, especially in the elimination of artifacts in low-dose CT images while preserving the structure.

The rest of this paper is organized as follows. In Section 2, we propose novel networks for noise estimation and super-resolution, together with the materials for implementing our experiments. In Section 3, the proposed image quality enhancement method is compared with the state-of-the-art methods by recovering low-resolution noisy images in a public dataset. Finally, we conclude our proposed work in Section 4.

## 2. Materials and Methods

### 2.1. Dense-Inception Network for Noise Estimation

As discussed in [33], the quantum and system noise in a CT image can be simulated as an additive noise that follows a Gaussian probability density function:(1)$$p\left(x\right)=\frac{1}{\sqrt{2\pi }\sigma }e^{-\frac{x^{2}}{2\sigma^{2}}},$$
where σ is the standard deviation and x is the random variable. The noise is distributed all over the image following the same pattern, so it can be considered as part of the image itself and its distributed pattern can be considered as a low-level characteristic of the image, similar to the edges and textures in the image. Therefore, the degradation network should not be designed too deep, or the features will be easily filtered out during the process of network transmission and overfitting is likely to happen. In this section, we take advantage of the inception structure and dense connection to classify the noise levels in medical images. An inception structure can extract the noise features of medical images in different receptive fields, while a dense connection can avoid the feature vanishing problem when the network is deeper.

#### 2.1.1. Overall Network Structure

Figure 2 shows the whole structure of the noise estimation framework. It mainly consists of the following parts:Pre-processing module: We used of a convolutional layer with 96 channels as the pre-processing layer of the degradation estimation network to convert the input image into the 96-dimensional feature space. A batch normalization (BN) layer and a rectified linear units (ReLU) layer [34] were followed to avoid the problem of gradient vanishing. After that, a max-pooling layer was applied to reduce the size of each feature map so as to decrease the amount of calculation.Inception module: Three inception-residual blocks were connected in series to compose the inception modules for the noisy image feature extraction. The inception module uses feature maps from the pre-processing module as input and processes them using receptive fields with different sizes. It guarantees that the image features with multiple scales can be preserved as much as possible, because the inception structure consists of convolution layers with different kernel sizes.Dense connection: The theory of skip connection proposed in Dense-Net [32] was used to improve the performance of our proposed degradation estimation network. The skip connections achieve the reuse of image features that are not processed by the corresponding inception-residual modules, so that it can reduce the disappearance of image noise features in the processing of feature transmission and fusion. Moreover, bottle necks were adopted in the dense connection, which keep the parameters at a reasonable amount and avoids overfitting.Global average-pooling and fully connected layer: After the inception module with dense connection, we applied average-pooling, whose kernel size was the size of feature maps to get the mean of each feature map, leading to a further reduction of the parameters and inhibition of over-fitting. Then, a full connection layer was used to output the probability of the noise level classification through the soft-max layer. The type with the largest probability was considered as the estimation of image degradation.

In the next two parts in this section, we will introduce the essential steps, that is, inception module and dense connection, in detail.

#### 2.1.2. Inception Structure

In our work, we treated noise and blurring as parts of the given image. The problem of degradation estimation could be consequently converted to the problem of classifying the types of degraded images. Given that the noise and blurring are distributed in both global image and local neighborhoods, we hoped to extract their features using different receptive fields. Moreover, these degradation features are low-level features, so we needed to use as few parameters as possible.

Figure 3 shows the inception-residual block we used in our proposed network. By mimicking the mechanism of how human beings observe a given image, the convolution layers with different sizes are parallel concatenated in one block, which means the same input layer is filtered by convolution kernels with different sizes and results from these different channels are concatenated together, so the receptive features of the noise or blurring are extracted from the image more comprehensively. Before being connected to the input of the certain inception-residual block, a 1 × 1 convolution layer is applied to reduce the dimensionality of the feature maps so that the whole block outputs less parameters while preserving most information. As shown in Figure 2, three inception-residual blocks are connected in series to extract the image features at a reasonable depth to improve the expression ability of the network.

#### 2.1.3. Dense Connection of Inception Modules

The image features are very likely lost during the forward transmission of the network, resulting in incomplete features of degradation in the final output. To solve this problem, we used the skip connections proposed in the Dense-Net [32]. It connects each layer to all the others in a feedforward manner, which means that the feature maps from all the previous layers are used as input of the current layer and the feature map from the current layer serves as the input of all the subsequent layers.

As shown in Figure 4, the three inception-residual modules are connected in the dense manner. Assuming the output of layer, n, is $X_{n}$, the relationship between them can be represented as follows:(2)$$X_{n}=H_{n}\left(X_{0},X_{1},\ldots ,X_{n}\right),$$
where $X_{n}$ is the feature map from the n-th layer, and $H_{n}()$ denotes the convolution operation of the n-th layer. The input of the $X_{n}$ layer is the concatenated features of those from previous *n* − 1 layers.

In medical images, degradation features, especially noise features, very likely disappear when the network is deep since those “non-structure” features are shallow in the image. Dense connection supports the reuse of features, meaning the necessary features are always preserved in the input of the next layer, so the problem of feature vanishing is avoided. Figure 5 shows the visualization of some feature maps extracted from the last short-cut connection layer (adding the last 1 × 1 convolutional layer and the input) of the third inception block. It can obviously be considered that a large amount of noise and blurring features are extracted from the proposed inception-residual modules with dense connections. Note that we adopted the bottleneck layer in the dense structure to further reduce the number of parameters and avoid over-fitting.

### 2.2. Dense-Residual-Inception Network for Image Quality Enhancement

Following the theory proposed in SRMD [22], we propose a dense-residual-inception network by taking the degraded image together with the noise level estimated from the network proposed in Section 2 as input to reconstruct a high-resolution image with less noise.

#### 2.2.1. Concatenation of the Degraded Image and Degradation Maps

An important characteristic of the super-resolution network for multiple degradations is to take both the degraded image and degradation maps as the input of the network. As discussed in [22], the given blurring kernel is vectorized, projected onto a t-dimension space, and then stretched into a tensor of a size, *W* × *H* × (*t* + 1), with the noise level, forming the degradation maps. Then, the degraded image and degradation maps are fused as the input of the image reconstruction network. Note that in our work, the noise level was adaptively estimated, other than being assumed as prior knowledge. Then, the degradation maps were concatenated with the degraded image and processed together by the subsequent network.

#### 2.2.2. Residual-Dense-Inception Network

As shown in Figure 6, a novel network, the so-called residual-dense-inception network, is proposed to recover the image from low-resolution, noisy, and blurred interruption. It mainly consists of four different parts, including the pre-processing module, residual-dense-inception module, sub-pixel convolution module, and reconstruction module. We will introduce them in detail as follows. Pre-processing moduleSimilar to that used in the degradation estimation network, here, the pre-processing module was also adopted to roughly extract shallow image features. However, since the input can be considered as an image block with multiple layers that contain a degraded image, pre-set blurring kernel, and noise map, the pre-processing module in the image reconstruction network plays the role of relating the image features and those degradation factors together. The input image block was mapped from the image space to the feature space, where the 128-dimensional feature maps were composed of 128 dimensions of extracted features.Residual-dense-inception moduleFigure 7 shows our proposed residual-dense-inception block to extract global dense features from input shallow features. To make full use of the multiple shallow features extracted from the input image block, we adopted the residual-dense block proposed in [24] as the basis of this module. The skip connections in the residual-dense block preserves the features from all the previous layers. The residual connection combines the shallow features and deep features so that the edges, textures, and tiny structures in the degraded image can be recovered better. Furthermore, we embedded the inception block into all the skip connection and residual connection routines to eliminate the noise information from the features because it functions little in the following sub-pixel convolution and reconstruction modules. Figure 8 shows the inception block we applied in our network, which was originally proposed in [31]. Three residual-dense-inception blocks were connected in series to get the accumulative dense features.Sub-pixel convolution moduleAfter the pre-processing module and the residual-dense-inception module, the size of the feature map was 40 × 40, which is much lower than our target image dimension 128 × *r*^2^ (*r* is the scale factor). To increase the resolution of the feature map, each pixel of *p*^2^ channels were re-arranged into an *r* × *r* image block corresponding to the sub-region with size *r* × *r* in the target high resolution feature map. Therefore, the feature map of 128 × *r*^2^ × *H* × *W* was re-arranged into a feature map with a higher dimensionality, 128 × *rH* × *rW*. Different from the traditional interpolation, the *rH* × *rW* region includes degraded information more adaptively, leading to a better super-resolution effect.Reconstruction moduleThe high-resolution feature maps processed by the sub-pixel convolution module were fully connected to form a full image by a convolution layer and a sigmoid layer. In our work, the 3 × 3 convolution kernel was defined to map the 128-channel feature maps into a single-channel grayscale image with higher resolution and lower degradation.

#### 2.2.3. Joint Loss of the Residual-Dense-Inception Network

Both the pixel-wise loss and the perceptual loss were used as the objective of the enhancement network for parameter optimization. The joint loss function was defined as follows:(3)$$L_{joint}=L_{pix}+\mu L_{perc},$$
where $L_{joint}$ is the joint loss, $L_{pix}$ and $L_{perc}$ represent the pixel-wise loss and perceptual loss, respectively, and μ is the loss normalization coefficient.

For the pixel-wise loss part, we adopted the traditional mean square error (MSE) loss, which can be mathematically expressed as follows:(4)$$L_{MSE}=\frac{1}{WH}{\left\|F\left(y\right)-x\right\|}^{2},$$
where y is the low-resolution noisy image, x is the high-quality real image, F(y) is the reconstructed image, and W and H are the width and height of the image, respectively.

For the perceptual loss part, we processed both the reconstructed image and the ground truth image by the VGG16 network [30], which is well-known for its ability in mimicking how human beings observe and understand the image to extract the high-level information [35], so called perceptual features, from the input image. We extracted the eighth layer output of the VGG16 network as the perceptual feature vectors of the reconstructed image and the target image, respectively. The perceptual loss was consequently defined as the Euclidean distance between these two perceptual feature vectors:(5)$$L_{Perc}=\frac{1}{W_{i}H_{i}}{\left\|\phi_{i}\left(F\left(y\right)\right)-\phi_{i}\left(x\right)\right\|}^{2},$$
where $W_{i}$ and $H_{i}$ denote the width and height of the convolution output of layer i of the VGG16 network, and $\phi_{i}$ is the *i*th layer that is used for feature extraction. We empirically set *i* as 8 because experiments show that the eighth layer of the VGG16 network can provide the best effect.

The flow of obtaining the joint loss and optimizing the network with it is shown in Figure 9. Both the pixel-wise differences and the structure perceptual differences between the reconstructed image and the ground truth image are taken into account to optimize the parameters of the reconstruction network. By minimizing the joint loss, the network learns the differences at the pixel and semantical level simultaneously, which enables the reconstruction network to retain more details and edge information while de-noising and achieving super-resolution, and finally produces clearer reconstruction images.

### 2.3. Experimental Materials

#### 2.3.1. Dataset

In our experiment, we used a public clinical data collection consisting of images and associated data acquired from the Applied Proteogenomics Organizational Learning and Outcomes (APOLLO) network, which is authorized by the cancer imaging archive (TCIA) [36], for the training and evaluation of the proposed networks. The dataset contains 1600 512 × 512 lung CT images, and we simulated the training and testing sets for the degradation estimation network and image quality enhancement network in two different ways:Dataset for the degradation estimation network: In total, 42,000 image patches with the size of 100 × 100 were randomly cropped from the 1600 lung CT images. One integer was randomly generated within a variation range of 0 to 18 and used as the variance of the Gaussian noise added to every image patch. Note that we treated each image patch with random noise as an individual noisy image and the variances of different additive noise were used as the labels of the images for classification. As a result, we obtained 42,000 degraded images with the size of 100 × 100, of which 40,000 were used as the training set and 2000 were used as the testing set. Figure 10 shows some sampled degraded image patches with different noise levels. The noise level equals the variance of the Gaussian noise added to the image patch. For example, noise level 8 means the variance of the Gaussian noise added to the image patch is 8.Dataset for the image quality enhancement network: All 1600 lung CT images were corrupted with random Gaussian white noise (5, 10, 15) and down-sampled to 256 × 256 images. Then, 1400 pairs of degraded and target images were used as the training set, while the other 200 pairs of images were used as the testing set. Figure 11 shows some sampled degraded images with noise and low-resolution.

#### 2.3.2. Comparator Methods

For comparison, we applied the state-of-the-art methods to our datasets. The comparator methods for the noise estimation network included the filter-based noise variance estimation [37], automatic noise parameter estimation (ANPE) [38], structure-oriented-based approach (SOBA) [39], and principal component analysis-based approach (PCA) [40]. Note that their parameters are the optimal parameters in the original paper. For the image quality enhancement, we compared our method with the following methods: The bicubic interpolation method [13], and several deep learning methods, including the super-resolution network for multiple degradations (SRMD) [22], the super-resolution using conditional GAN (SRCGAN) [41], the (RED-CNN) [42], and the residual dense network (RDN) [24].

### 2.4. Parameter Setting

For the noise estimation network, the cross-entropy loss was adopted as the loss function. The regularization coefficient was set to 1 × 10^4^. The network was trained by the Adam optimizer, and the initial learning rate was set to 0.001; β1 and β2 were set to 0.9 and 0.999, respectively. The batch size of training was set to 40, and the total epoch was 40.

For the image quality enhancement network, the input used in the training was a block with the size of 40 × 40, and the kernel size of each convolution layer was 3 × 3. Except the convolution layer after the residual-dense-inception module, which consisted of 128 × *r*^2^ (*r* is the scale factor) filter banks, the other convolution layers were all composed of 128 filter banks. The batch-size was set to 20 and the Adam optimizer was adopted for network training with the initial learning rate set as 0.01. After 200,000 steps of iteration, the training stopped.

All training and testing were carried out under the Pytorch framework on an Intel Core i7-4790K 4.0 GHz PC with 16 G RAM and an NVIDIA TITAN XP GPU with 12 G RAM.

### 2.5. Experiment Implementation

The experiments were implemented as follows:The testing noisy image patches were processed to provide the estimated noise level by the proposed noise estimation network, as well as by the comparators.The accuracy and the confusion rate of the classification of noise levels by different methods were calculated to evaluate the performance of our proposed method quantitatively.The testing of noisy low-resolution images were processed to provide clean high-resolution images by the proposed image quality enhancement network with the noise level generated from the well-trained noise estimation network, as well as by the comparator methods.The peak signal to noise ratio (PSNR) [43] and structured similarity index (SSIM) [43] were calculated by comparing the resulting reconstructed images with the ground truth images, leading to a quantitative evaluation of the proposed method.

## 3. Results and Discussion

In this section, we apply our proposed method to a public lung CT image dataset and evaluate its performance on degradation estimation and the subsequent quality enhancement compared to the state-of-the-art methods. Details of the experimental implementation and experimental results are discussed as follows.

### 3.1. Performance on Noise Estimation

Figure 12 shows the accuracy curves of the classification of noisy image patches into different noise levels with different estimation methods. Table 1 shows the confusion rates of the confusion matrices generated by these estimation methods. All methods illustrate their capabilities to estimate the noise level with reasonable accuracies. However, the filter-based method [37] provides poor performance when the noise level is high. Additionally, the high confusion rate (0.25) of the filter-based method shows that it is likely that it recognizes the very high-level noise as “structures” by mistake. Compared with the filter-based method, the ANPE [38] and SOBA [39] can improve noise estimation accuracy, especially when the noise level is high. However, the estimation error still gradually increases while the noise level increases. Additionally, the confusion rate of the high-level noise estimation shows an unstable trend. The PCA-based method [40] is better than the previous methods in noise estimation. However, it pays the price of under-estimating the noise level; that is, it classifies the low-level noise as little noise but edge textures in the image, resulting in high confusion rates for the low-level noise. The proposed method can extract noise information on the basis of the whole image, so the estimation of noise level is more accurate with lower confusion rates.

To further evaluate the performance of the proposed method, Table 2 illustrates the confusion matrix, precision, and recall of the classification of different noise levels with the proposed method explicitly. In the cases of category 1 and 2, which correspond to a relatively low noise level, the proposed method provides precision values of 0.86 and 0.955, recall values of 0.96 and 0.84, and F1-measure values of 0.91 and 0.89, respectively. It is because the noise features are not obvious enough when there is little noise, and it is easy to confuse the noise level of category 1 and 2. However, when the noise level increases, the noise features get more obvious, so the proposed method, with the ability to differentiate noisy images with different visual features, can retain high precision and recall values. The performance is especially good in the case of noise level 4 and 18, with precision, recall, and F1-measure values of 1.00, 1.00, and 1.00, respectively. One reason for this phenomenon is that noise level 4 and 18 are two watersheds between “very little noise”, “some noise”, and “much noise”, which are obviously different from their adjacent noise levels from human perception and their features can be extracted and recognized well by the proposed network. Another reason for this is that the amount of testing samples is relatively small. We will implement the network on a larger database in the future. Moreover, the errors are distributed close to the correct noise level, which means the error only occurs when the noise levels are quite close to each other.

### 3.2. Performance on Image Super-Resolution

#### 3.2.1. Examination of Design Strategies

Figure 13 illustrates the performance of different loss functions in restraining the proposed image super-resolution network, including MSE-loss, perceptual-loss, and MSE-perceptual-loss. Using only MSE as a loss function results in the loss of details of the structures in the image, while using the perceptual loss individually leads to blurring and incomplete elimination of noise. In contrast, combining the MSE and perceptual loss can achieve a good balance in the removal of noise and the preservation of structures. The PSNR and SSIM results shown in Table 3 confirm this observation.

#### 3.2.2. Comparisons with Other Models

Figure 14 shows the result of enhancing the quality of images corrupted by noise and low-resolution through different methods. The bicubic interpolation method provides the worst performance, with much remaining noise and artifacts along the edges in the high-resolution image. This is mainly because the bicubic method simply makes use of the intensities in the neighborhood without taking the structure information into account. When the noise is distributed around the image, the noise intensities are also calculated by the interpolation function in the high-resolution image, resulting in the artifacts and blurring as shown in Figure 14(c1–c3). The deep learning methods perform better than the classic interpolation method because they make use of the visual features of the image so that the noise can be eliminated. However, the SRMD needs to know the noise level as prior knowledge. When the actual noise is different from the assumed noise, the performance is not as convincing. In our experiment, the noise level is unknown and was set by the user blindly, so it can be considered in Figure 14(d1–d3) that the structures, edges, or other details are not reconstructed in the resulted image. As shown in Figure 14(e1–e3), the SRCGAN can recover the structures in the medical image well, but some noise cannot be eliminated well because the GAN may consider this noise as tiny structures by mistake. In contrast, as shown in Figure 14(f1–f3), the RED-CNN can provide a good performance in the removal of noise because the U-net structure can filter out the noise features very well. However, it will over-smooth small edges in the image because it lacks the ability to preserve detailed information and perceptual features. As shown in Figure 14(g1–g3), the RDN method performs better than the previous methods because the residual-dense block can extract and reuse the image features better than the simple convolution layer used in the SRMD, SRCGAN, and RED-CNN. However, noise remains in the high-resolution image due to the ability of the skip connection to transfer noise features to the last fully connection layer as well as the structure features. Moreover, the RDN also needs to have prior knowledge of the noise level in the image, which makes it unable to adapt to a blind case. Our proposed method takes advantage of the RDN method and improves it by embedding an inception block into the skip connection routines. Moreover, it makes use of the noise estimation network to automatically calculate the noise level and takes that as the input of the quality enhancement network. Therefore, as shown in Figure 14(h1–h3), the proposed framework provides superior performance on noise removal and detail preservation during the super-resolution.

Table 4 shows the values of PSNR and SSIM using different methods to reconstruct 200 images in the testing set with different noise levels. We can see that the proposed method provides the highest PSNR and SSIM values in cases of the most noise levels, which confirms our pervious qualitative observations.

Figure 15 shows the absolute differences between the images restored by different methods and the ground truth image. It demonstrates that the proposed method can eliminate noise the most while removing the anatomical structures to the least extent.

## 4. Conclusions

We proposed a uniform framework with a novel architecture for image de-noising and super-resolution simultaneously. A dense-inception network was proposed by integrating an inception structure and dense skip connection to estimate the noise level. Another novel network was proposed by embedding inception blocks into skip connections of residual-dense blocks for simultaneous noise removal and super-resolution. The two proposed networks were incorporated by using the output of the dense-inception network as the input of the residual-dense-inception network for image quality enhancement. The mean square error (MSE)-based pixel-wise loss and the VGG-16-based perceptual loss were combined together as the loss to optimize the whole network. The experimental results on the public dataset proved that the proposed method can effectively achieve noise removal and super-resolution in one single framework. Therefore, the proposed method outperformed the state-of-the-art methods in visual effects and quantitative measurements.

In the future, we will follow the proposed medical image quality enhancement strategy that estimates prior knowledge first and then removes noise and increases the resolution simultaneously. We will improve the noise estimation part by considering noise with more types or a more complicated distribution, especially streak artifacts in low-dose CT images. For noise removal and super-resolution, we will try to propose a network with convolutional blocks that can extract features closer to the knowledge from clinical experts. For the training of the whole network, we may make use of the multi-task learning strategy to optimize the parameters of both the noise estimation part and the quality enhancement part simultaneously. Moreover, we will evaluate our method on some other databases with a larger amount of images for a more comprehensive evaluation.

## Figures and Tables

**Figure 1 sensors-19-03348-f001:**
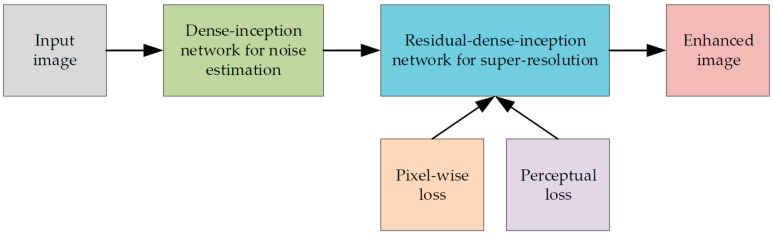
The framework of the proposed image quality enhancement method.

**Figure 2 sensors-19-03348-f002:**
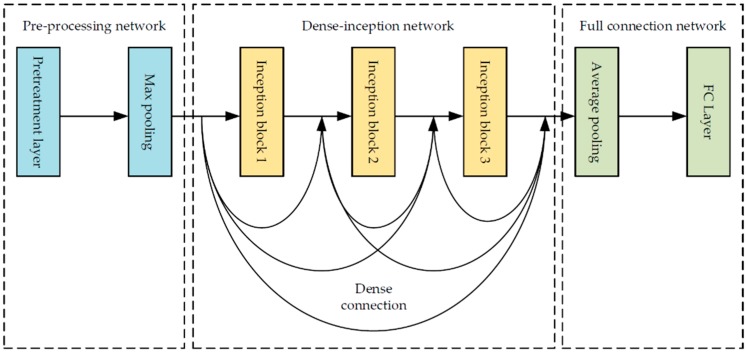
The whole structure of the noise estimation framework.

**Figure 3 sensors-19-03348-f003:**
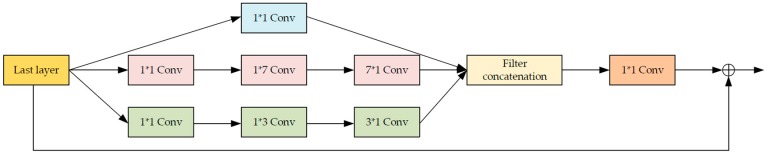
The inception-residual block used in the proposed network.

**Figure 4 sensors-19-03348-f004:**
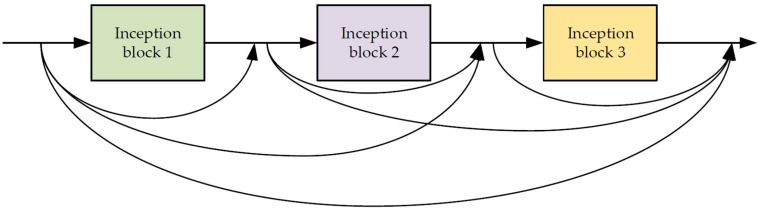
The dense connection of three inception blocks.

**Figure 5 sensors-19-03348-f005:**
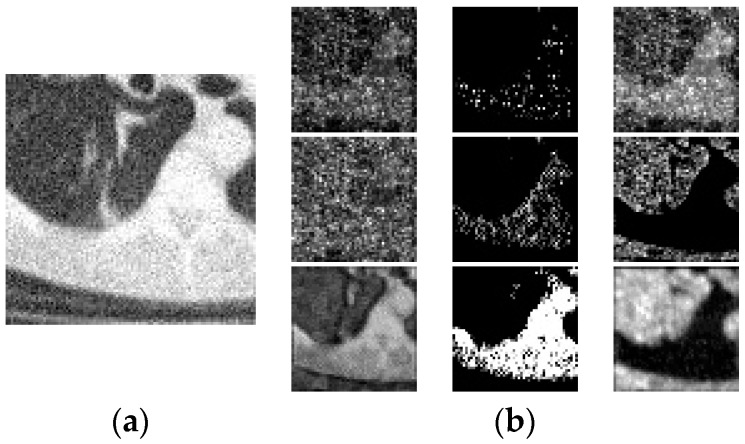
Visualization of feature maps extracted from the proposed noise estimation network when taking the noisy image as input. (**a**) is the input image patch, (**b**) are several feature maps as the visualization of the last short-cut connection layer (adding the last 1 × 1 convolutional layer and the input) of the third inception block of the proposed noise estimation network.

**Figure 6 sensors-19-03348-f006:**
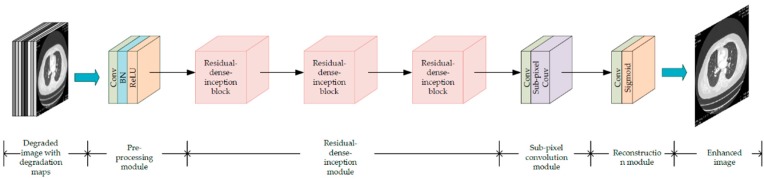
The structure of the proposed residual-dense-inception network for image super-resolution with multiple degradations.

**Figure 7 sensors-19-03348-f007:**
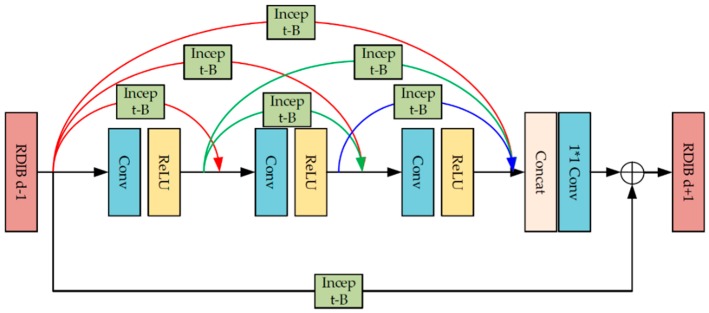
The architecture of the proposed residual-dense-inception block.

**Figure 8 sensors-19-03348-f008:**
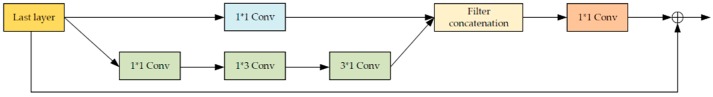
The inception block used in the residual-dense-inception block shown in Figure 7.

**Figure 9 sensors-19-03348-f009:**
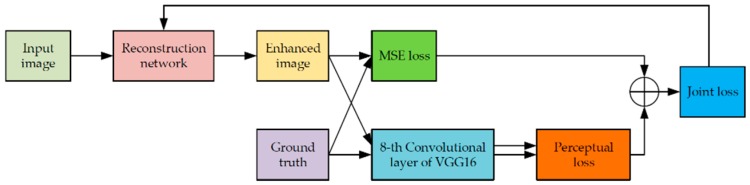
The flow of calculating the joint loss for optimizing the parameters of the reconstruction network.

**Figure 10 sensors-19-03348-f010:**
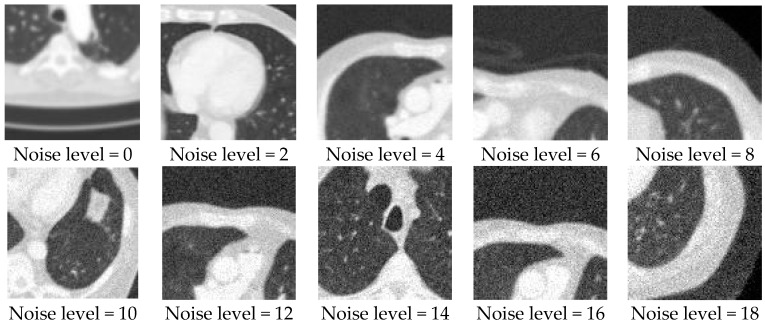
Sampled image patches with different noise levels as the dataset for the image noise estimation network.

**Figure 11 sensors-19-03348-f011:**
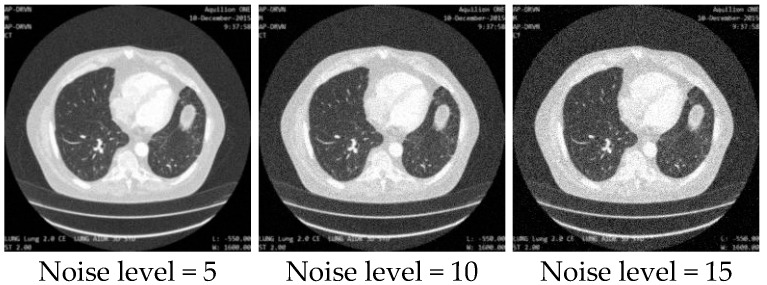
Sampled images with low resolution and different noise levels as the dataset for the image quality enhancement network.

**Figure 12 sensors-19-03348-f012:**
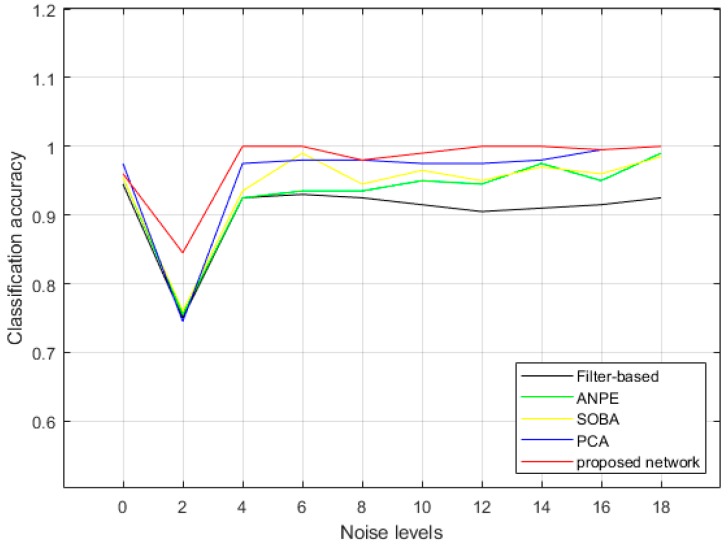
The accuracy curves of the classification of noisy image patches into different noise levels with different estimation methods.

**Figure 13 sensors-19-03348-f013:**
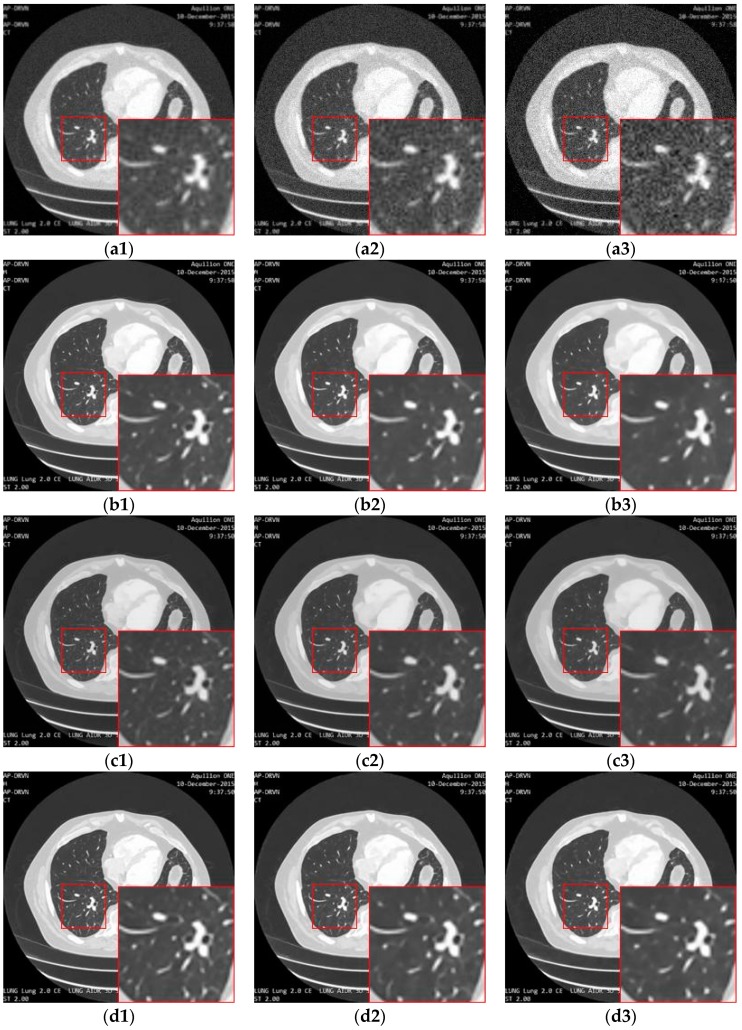
The result of enhancing the quality of images corrupted by noise with different levels and low-resolution through the proposed network with different loss strategies. (**a1**), (**a2**) and (**a3**) are the input images with noise level 5, 10 and 15. (**b1**), (**b2**) and (**b3**) are the ground truth images. (**c1**), (**c2**) and (**c3**) are the images (**a1**), (**a2**) and (**a3**) processed by the proposed network with MSE loss. (**d1**), (**d2**) and (**d3**) are the images (**a1**), (**a2**) and (**a3**) processed by the proposed network with the MSE-perceptual joint loss.

**Figure 14 sensors-19-03348-f014:**
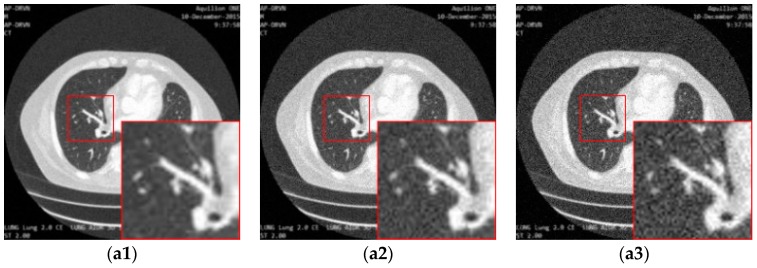
The result of enhancement of the quality of images corrupted by noise with different levels and low-resolution through different methods. (**a1**), (**a2**) and (**a3**) are the input images with noise level 5, 10 and 15. (**b1**), (**b2**) and (**b3**) are the ground truth image. (**c1**), (**c2**) and (**c3**) are the images (**a1**), (**a2**) and (**a3**) processed by the bicubic method. (**d1**), (**d2**) and (**d3**) are the images (**a1**), (**a2**) and (**a3**) processed by the SRMD method. (**e1**), (**e2**) and (**e3**) are the images (**a1**), (**a2**) and (**a3**) processed by the SRCGAN method. (**f1**), (**f2**) and (**f3**) are the images (**a1**), (**a2**) and (**a3**) processed by the RED-CNN method. (**g1**), (**g2**) and (**g3**) are the images (**a1**), (**a2**) and (**a3**) processed by the RDN method. (**h1**), (**h2**) and (**h3**) are the images (**a1**), (**a2**) and (**a3**) processed by the proposed method.

**Figure 15 sensors-19-03348-f015:**
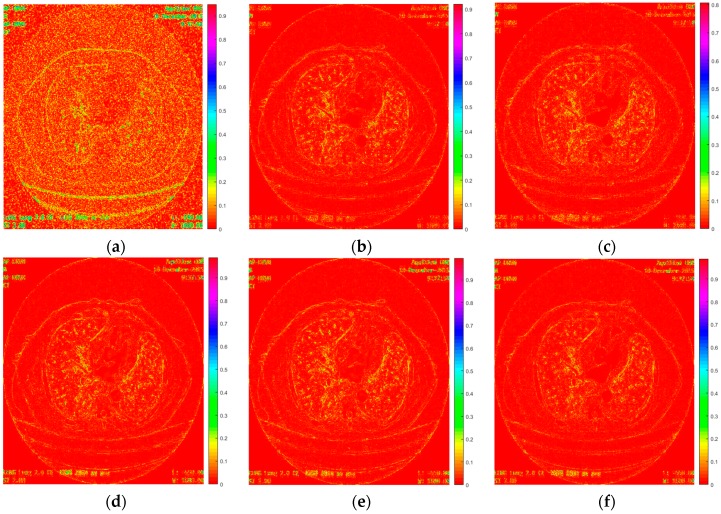
Absolute differences between the images (noise level = 20) restored by different methods and the ground truth image. (**a**) is the bicubic interpolation, (**b**) is the SRMD, (**c**) is the SRCGAN, (**d**) is the RED-CNN, (**e**) is the RDN, and (**f**) is the proposed method.

**Table 1 sensors-19-03348-t001:** The confusion rates of the confusion matrices generated by the classification of noisy image patches into different noise levels with different estimation methods.

Noise Level	0	2	4	6	8	10	12	14	16	18
Fiter-based	0.055	0.250	0.075	0.070	0.075	0.085	0.095	0.090	0.085	0.075
ANPE	0.045	0.245	0.075	0.065	0.065	0.050	0.055	0.025	0.050	0.010
SOBA	0.045	0.240	0.065	0.010	0.055	0.035	0.050	0.030	0.040	0.015
PCA	0.025	0.255	0.025	0.020	0.020	0.025	0.025	0.020	0.005	0.000
proposed	0.040	0.160	0.000	0.000	0.020	0.010	0.000	0.000	0.005	0.000

**Table 2 sensors-19-03348-t002:** The confusion matrix, precision, and recall of classifying 10 noise levels by the proposed network.

Predicted											
		0	2	4	6	8	10	12	14	16	18
Actual	0	192	8	0	0	0	0	0	0	0	0
	2	31	169	0	0	0	0	0	0	0	0
	4	0	0	200	0	0	0	0	0	0	0
	6	0	0	0	200	0	0	0	0	0	0
	8	0	0	0	3	196	1	0	0	0	0
	10	0	0	0	0	0	198	2	0	0	0
	12	0	0	0	0	0	0	200	0	0	0
	14	0	0	0	0	0	0	0	200	0	0
	16	0	0	0	0	0	0	0	1	199	0
	18	0	0	0	0	0	0	0	0	0	200
P		0.86	0.955	1.00	0.985	1.00	0.995	0.99	0.995	1.00	1.00
R		0.96	0.84	1.00	1.00	0.98	0.99	1.00	1.00	0.995	1.00
F1-measure		0.91	0.89	1.00	0.99	0.99	0.99	0.99	0.99	0.99	1.00

**Table 3 sensors-19-03348-t003:** Quantitative results (mean ± std of PSNR and SSIM) associated with different combinations of losses of the proposed network for enhancing the qualities of the images with different noise levels.

Noise Level	5		10		15	
	PSNR	SSIM	PSNR	SSIM	PSNR	SSIM
MSE	29.12 ± 0.18	0.84 ± 0.01	28.53 ± 0.23	0.83 ± 0.01	28.09 ± 0.15	0.80 ± 0.01
Perceptual	28.09 ± 0.15	0.85 ± 0.01	27.85 ± 0.27	0.84 ± 0.01	27.12 ± 0.57	0.83 ± 0.01
Perceptual-MSE	30.79 ± 0.25	0.88 ± 0.01	30.23 ± 0.41	0.86 ± 0.01	29.25 ± 0.42	0.84 ± 0.01

**Table 4 sensors-19-03348-t004:** Quantitative results (mean ± std of PSNR and SSIM) associated with different algorithms for enhancement of the qualities of images with different noise levels.

Noise Level	5		10		15	
	PSNR	SSIM	PSNR	SSIM	PSNR	SSIM
Bicubic	23.53 ± 0.17	0.70 ± 0.01	23.22 ± 0.15	0.53 ± 0.01	22.74 ± 0.15	0.40 ± 0.01
SRMD	28.89 ± 0.16	0.85 ± 0.01	28.62 ± 0.20	0.83 ± 0.01	28.13 ± 0.22	0.82 ± 0.01
SRCGAN	28.38 ± 0.11	0.86 ± 0.01	27.95 ± 0.26	0.85 ± 0.01	27.78 ± 0.13	0.83 ± 0.01
RED-CNN	29.14 ± 0.18	0.85 ± 0.01	28.71 ± 0.23	0.83 ± 0.01	28.54 ± 0.17	0.82 ± 0.01
RDN	29.27 ± 0.21	0.87 ± 0.01	28.89 ± 0.17	0.86 ± 0.01	28.67 ± 0.55	0.83 ± 0.01
proposed	30.79 ± 0.25	0.88 ± 0.01	30.23 ± 0.41	0.86 ± 0.01	29.25 ± 0.42	0.84 ± 0.01

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
