# Peer review of "Computed Tomography (CT) Image Quality Enhancement via a Uniform Framework Integrating Noise Estimation and Super-Resolution Networks"

_sensors, 2019, doi:10.3390/s19153348_

Round 1

Reviewer 1 Report

This paper describes a method to enhance medical image quality by removing noise using a CNN-based approach.  In short, a network is used to learn and estimate the level of noise in the image and then the noise is removed to produce a cleaner image.  The TCIA pubic dataset is used for evaluation and results are shown that clean up noise in the image.

The positives for the paper are that it is thorough, the methods are described well, and prior art is reasonably considered.  I think there has been a lot of work in image enhancement using deep learning both inside and outside the medical imaging community and only a very small subset is considered here.  I think more work should be considered.  Also the figures are clear and when possible, the authors try to give intuitive reasoning for algorithmic choices in their architecture.

On the flip side, I do think this paper needs a lot of work before it is published.  First, the English needs some fixing.  Some sentences don't make sense to me, for example, the first sentence in 2.1: "Since the noise is distributed all over the image, the noise feature is shallow, global features".  

Here are a few other questions/concerns/suggestions:

- in general, a lot of claims are made that are not cited.  While they may be true, we still must cite them to prove them.  In the abstract it says that "the image quality is usually interrupted during the process of imaging" -- does this just mean there is noise or the imaging process itself is interrupted?  If so, we need some proof of this.  In the introduction, it says "noisy, low-resolution images that cannot be used for those subsequent steps".  I think this is a bit overstating the problem otherwise CTs wouldn't be useful in medicine at all.  I do think its an issue so we need a better description of how big the problem actually is.  These are just basics of motivating the problem to me (in general).

- I think the paper lumps in all types of medical imaging but then just seems to address CT.  I think this is a bit too broad.  Different imaging modalities have different issues.

- when you put multiple citations, instead of, for example, [25,26,27,28], do [25-28]. I saw this at least twice.

- I think the architecture itself isn't really that novel.  I think that deep learning has been explored and published so much to date that mixing and matching blocks from one architecture into another isn't really moving the field forward--it's engineering.  It is said that "the inception structure can extract features of noise under multiple receptive fields...".  All CNN architectures are multi-resolution.  They all extract features, down-sample, extract features, down-sample, etc.  So this isn't a claimed novelty of this approach.  

- There has been A LOT of work using U-NET style networks that have been very successful.  Also GAN approaches have shown superior results as well for image enhancement.  I think there needs to be comparisons against these types of approaches as well as reasoning why the proposed approach is better than those.

- figure 5 shows features extracted by different layers. Where in the network?  it would be nice to know how deep these are taken from.

- where did this perceptual loss come from?  another paper?  or proposed here?  I dont see why this is "perceptual".  

- the authors should do ablation studies to show the MSE, perceptual, and MSE+perceptual to show that the combination is actually better.

- I think the experiments are odd.  the overall dataset is small to be training deep networks from scratch (~1600).  The authors claim early on that global noise is important to be considered and that this approach learns to model both global and local noise through inception blocks and dense networks.  But then very small patches are used as input.  I would think these are not containing any global information relative to the original image?

- am I right in assuming you are simulating the noise by adding the Gaussian white noise?  if so, this seems too simple to me to extract.  At least it should be shown (or cited) that gaussian noise accurately models real noise that is observed, such as transmission noise, etc.

- I didn't notice any deep learning methods as comparison.  I think that needs to be in there.  And I really think GANs and/or U-Nets need to be considered.

- for the noise estimation results, anytime I see precision = recall = 1.0 I get suspicious.  Either the problem was too simple or some over-fitting is occurring.  In either case I would think this doesn't really describe the results accurately.  I'm curious how that happens.  Maybe this is what I said earlier that Gaussian noise is too simple and easy to learn.

- how do we define "noise level"?  it shows that levels 1 and 2 don't work well and it is suggested that this is b/c the noise is so subtle.  But I have a hard time mapping in my mind what level 1 and level 2 mean with respect to the simulated noise?

- for the final results, b/c we are starting from only learning simulated noise, do we know if any real anatomical structures get removed as well?  is there a way to understand how much of the real noise is removed versus areas of the image that may not be noise?

Author Response

Dear Reviewer,

Thank you very much for reviewing our manuscript, “Computed Tomography (CT) image quality enhancement via a uniform framework integrating noise estimation and super-resolution networks”, for the publication in the awesome journal “Sensors”. Your comments for the manuscript were highly insightful and enabled us to improve the quality of our manuscript. All the revisions we have made according to your comments are marked by red color in the revised manuscript and I will address our response to each comment below.

1. First, the English needs some fixing.  Some sentences don't make sense to me, for example, the first sentence in 2.1: "Since the noise is distributed all over the image, the noise feature is shallow, global features".

We have modified the sentence in 2.1 as “The noise is distributed all over the image following the same pattern, so it can be considered as part of the image itself and its distributed pattern can be considered as a low-level characteristic of the image, similarly as the edges, textures in the image”. What we want to express is that we treat the noisy image as an image other than image plus noise, so the noise can be considered as a low-level characteristics of the image, just as the edges or textures in the image.

2. In general, a lot of claims are made that are not cited.  While they may be true, we still must cite them to prove them.  In the abstract it says that "the image quality is usually interrupted during the process of imaging" -- does this just mean there is noise or the imaging process itself is interrupted?  If so, we need some proof of this.  In the introduction, it says "noisy, low-resolution images that cannot be used for those subsequent steps".  I think this is a bit overstating the problem otherwise CTs wouldn't be useful in medicine at all.  I do think its an issue so we need a better description of how big the problem actually is.  These are just basics of motivating the problem to me (in general).

We have clarified in both abstract and introduction part that the image quality is interrupted by two factors: 1) the noise during the process of imaging, and 2) the data compression during the process of storage and transmission, resulting in noisy and blurring image. And the degraded image will degrade the subsequent steps of CADs, including image segmentation, feature extraction and medical image classification.

3. I think the paper lumps in all types of medical imaging but then just seems to address CT.  I think this is a bit too broad.  Different imaging modalities have different issues.

We have modified the title of the manuscript to “Computed Tomography (CT) image quality enhancement via a uniform framework integrating noise estimation and super-resolution networks”, as well as the expression in abstract and introduction, so it has been clarified that our work focused on CT images.

4. When you put multiple citations, instead of, for example, [25,26,27,28], do [25-28]. I saw this at least twice.

We have modified the way of citing multiple works as you suggest in Line 97 and Line 99 in the introduction part.

5. I think the architecture itself isn't really that novel.  I think that deep learning has been explored and published so much to date that mixing and matching blocks from one architecture into another isn't really moving the field forward--it's engineering.  It is said that "the inception structure can extract features of noise under multiple receptive fields...".  All CNN architectures are multi-resolution.  They all extract features, down-sample, extract features, down-sample, etc.  So this isn't a claimed novelty of this approach.

We have modified the expressions from Line 114 to Line 126 in the Introduction part to clarify our novelty: We novelly proposed a uniform framework to estimate the noise level (variance) and use it as prior-knowledge for image super-resolution, so the reconstruction network can process the noisy, blur medical images with much less prior knowledge about noise than the state-of-the-art works.

6. There has been A LOT of work using U-NET style networks that have been very successful.  Also GAN approaches have shown superior results as well for image enhancement.  I think there needs to be comparisons against these types of approaches as well as reasoning why the proposed approach is better than those.

We have added a U-net style network and a GAN-style network in our experiments to show the ability of our proposed method. And we analyzed the experimental results of these two networks from Line 459 to Line 464. While the qualitative and quantitative results were added in Figure 14 and Table 4.

7. Figure 5 shows features extracted by different layers. Where in the network?  it would be nice to know how deep these are taken from.

We have modified the title of Figure 5, as well as the expression from 225-226, to clarify that the features were from the last short-cut connection layer (adding the last convolutional layer and the input) of the 3rd inception block.

8. Where did this perceptual loss come from?  another paper?  or proposed here?  I dont see why this is "perceptual".

We have added the expressions from Line 303 to Line 306, to make it clearer that the definition and the calculation of the perceptual loss. We process both the reconstructed image and the ground truth image by the VGG16 network, which is well known for its ability in mimicking how human beings observe and understand the image to extract the high-level information, so called perceptual features, from the input image.

9. The authors should do ablation studies to show the MSE, perceptual, and MSE+perceptual to show that the combination is actually better.

We have added the ablation studies of using MSE, perceptual and MSE+perceptual in Section 3.2.1. The qualitative results were shown in Figure 13 and the quantitative results were shown in Table 3.

10. I think the experiments are odd.  the overall dataset is small to be training deep networks from scratch (~1600).  The authors claim early on that global noise is important to be considered and that this approach learns to model both global and local noise through inception blocks and dense networks.  But then very small patches are used as input.  I would think these are not containing any global information relative to the original image?

We have changed the expressions from Line 335 to Line 340 to clarify the experiments for the noise estimation. And we stopped using the expression “global” because it may result in ambiguity. Our noise estimation network treat the noise as a part of image itself so no matter what the size of the image patch is, it can estimate the noise level in the image.

11. Am I right in assuming you are simulating the noise by adding the Gaussian white noise?  if so, this seems too simple to me to extract.  At least it should be shown (or cited) that gaussian noise accurately models real noise that is observed, such as transmission noise, etc.

Yes, in this work we assumed the noise in the image can be simulated as Gaussian white noise. We have added a citation in Line 146 to Line 149 showing that the quantum and system noise in CT image can be simulated as an additive noise that follows Gaussian probability density function.

12. I didn't notice any deep learning methods as comparison.  I think that needs to be in there.  And I really think GANs and/or U-Nets need to be considered.

We have added the GAN-style and U-Net-style networks in our experiments part as comparisons. And we clarified in Line 356 that the the super-resolution network for multiple degradations (SRMD) and the residual dense network (RDN) are both deep learning methods. We analyzed the experimental results of these two networks from Line 459 to Line 464. While the qualitative and quantitative results were added in Figure 14 and Table 4.

13. For the noise estimation results, anytime I see precision = recall = 1.0 I get suspicious.  Either the problem was too simple or some over-fitting is occurring.  In either case I would think this doesn't really describe the results accurately.  I'm curious how that happens.  Maybe this is what I said earlier that Gaussian noise is too simple and easy to learn.

We have added some explanation form Line 420 to Line 426 about the precision and recall results. One reason for this phenomenon is that noise level 4 and 18 are two watersheds between “very little noise”, “some noise”, and “much noise”, which are obviously different from their adjacent noise levels from human perception and their features can be extracted and recognized well by the proposed network. Another reason for this we should admit that the amount of testing samples is relatively small and we will implement the network on a larger database in the future.

14. How do we define "noise level"?  it shows that levels 1 and 2 don't work well and it is suggested that this is b/c the noise is so subtle.  But I have a hard time mapping in my mind what level 1 and level 2 mean with respect to the simulated noise?

We have added an explanation from Line 339 to Line 340 about the noise level. The noise level equals to the variance of Gaussian noise added to the image patch. For example, noise level 8 means the variance of the Gaussian noise added to the image patch is 8.

15. For the final results, b/c we are starting from only learning simulated noise, do we know if any real anatomical structures get removed as well?  is there a way to understand how much of the real noise is removed versus areas of the image that may not be noise?

We have shown the absolute differences between the images restored by different methods and the ground truth image in Figure 15 and we have discussed the results from Line 486 to Line 488. It demonstrated that the proposed method can eliminate the noise most while removing the anatomical structures to the least extent.

We thank you for your helpful feedback and we hope that the revisions in the manuscript and our accompanying responses will be sufficient to make our manuscript suitable for publication.

Best regards!

Yours,

Jianning Chi

Reviewer 2 Report

Please see the attached pdf file.

Author Response

Dear Reviewer,

Thank you very much for reviewing our manuscript, “Computed Tomography (CT) image quality enhancement via a uniform framework integrating noise estimation and super-resolution networks”, for the publication in the awesome journal “Sensors”. Your comments for the manuscript were highly insightful and enabled us to improve the quality of our manuscript. All the revisions we have made according to your comments are marked by blue color in the revised manuscript and I will address our response to each comment below.

1. F-measure and its evaluation are missing in Table 2.

We have added the F1-measure and its evaluation in Table 2.

2. The dense connection part in Fig. 2 is not the same as Fig. 4.

We have re-drawn the Figure 2 and the dense connection part is now same as that in Figure 4.

3. The name of the collection of TCIA applied in this manuscript is unclear.

We have added the name of the collection from Line 326 to Line 328 for a clearer expression. We use a public clinical data collection consisting of images and associated data acquired from the Applied Proteogenomics Organizational Learning and Outcomes (APOLLO) network, which is authorized by the cancer imaging archive (TCIA).

4. Long forms of abbreviations are required: PSNR and SSIM in Abstract.

We have added in Abstract part the long forms of abbreviations of PSNR and SSIM.

5. Future work is missing in Conclusions.

We have added the future work plan from Line 505 to Line 514 in the Conclusion part.

6. Italic or roman: Typefaces for symbols are not correctly used.

We have modified the typefaces for symbols in all the figures as Palatino Linotype as the Journal requirements of the manuscript.

We thank you for your helpful feedback and we hope that the revisions in the manuscript and our accompanying responses will be sufficient to make our manuscript suitable for publication.

Best regards!

Yours,

Jianning Chi

Round 2

Reviewer 1 Report

I will say I was very happy to see the revised version of this manuscript.  This is probably one of the bigger turn-arounds from an initial to second submission I've seen in a while.  The authors took the comments well and really improved the paper for me.  I was on the fence before with the initial version but feel much better about this one.  Just 1 small comment:

"

For the perceptual loss part, we process both the reconstructed image and the ground truth image by the VGG16 network, which is well known for its ability in mimicking how human beings observe and understand the image to extract the high-level information, so called perceptual features, from the input image.

"

-- this part is still a bit weak to me.  Are there 1-2 references you can put here, since it's so well-known?  If so that would strengthen this argument.  In principle I believe that these features mimic "something perceptual".  It's just a strong statement declaring it is all.  It's a nit-picky comment.

Author Response

Dear Reviewer,

Thank you very much for reviewing the revision of our manuscript, “Computed Tomography (CT) image quality enhancement via a uniform framework integrating noise estimation and super-resolution networks”, for the publication in the awesome journal “Sensors”. Your comments for the manuscript were helpful for us to improve the manuscript. The revision we have made according to your comment this time was marked by red color again in the revised manuscript and I will address our response to the comment below.

"For the perceptual loss part, we process both the reconstructed image and the ground truth image by the VGG16 network, which is well known for its ability in mimicking how human beings observe and understand the image to extract the high-level information, so called perceptual features, from the input image."

-- this part is still a bit weak to me.  Are there 1-2 references you can put here, since it's so well-known?  If so that would strengthen this argument.  In principle I believe that these features mimic "something perceptual".  It's just a strong statement declaring it is all.  It's a nit-picky comment.

We have added Reference [30] for the VGG-16 network and Reference [34] for the statement of the advantages of the VGG-16 (from Line 303 to Line 306). Note that in Reference [34] the “perceptual loss” was firstly claimed.

We thank you for the insightful feedback and we hope the revision in the manuscript and our accompanying response will make the statement clearer and the manuscript suitable for publication.

Best regards!

Yours,

Jianning Chi

Reviewer 2 Report

The manuscript addresses an interesting problem encountered in medical image quality enhancement for making de-noising and super-resolution by the uniform deep convolutional neural network. The reviewer recommends this revised manuscript to be published in the Sensors Journal.

Author Response

Dear Reviewer,

Thank you very much for reviewing and recommending the revision of our manuscript, “Computed Tomography (CT) image quality enhancement via a uniform framework integrating noise estimation and super-resolution networks”, for the publication in the awesome journal “Sensors”.

Best regards!

Yours,

Jianning Chi
